# The Importance of Emotional Intelligence in University Athletes: Analysis of Its Relationship with Anxiety

**DOI:** 10.3390/ijerph20054224

**Published:** 2023-02-27

**Authors:** Isabel Mercader-Rubio, Nieves Gutiérrez Ángel

**Affiliations:** Department of Psychology, Faculty of Education Sciences, Universidad de Almería, La Cañada, 04120 Almería, Spain

**Keywords:** emotional intelligence, pre-competitive anxiety, university students, physical education

## Abstract

In the field of sport psychology, research on emotional intelligence and its relationship with other psychological variables to determine how it affects the athlete’s performance is becoming more frequent and prevalent. Among these psychological variables, research in this field has focused on the evaluation of the influence of aspects such as motivation, leadership, self-concept, and anxiety. The main objective of this research is to analyze the levels of each of the dimensions of emotional intelligence (attention, clarity, and emotional regulation) and their relationship with each of the SCAT items to measure pre-competitive anxiety. To do so, we analyzed the influence that one psychological construct has on the other, in order to establish the type of relationships that are established between them. The design of this research corresponds to be transversal, observational, quantitative, and descriptive. The sample consisted of 165 students belonging to university degrees (bachelor’s and master’s) related to physical activity and sport sciences. The main finding of this study allows us to affirm the relationship between emotional intelligence and anxiety. This confirms the hypothesis that anxiety is an indispensable component of any competitive situation, and that neither the total absence of anxiety nor high levels of it leads to better sports performance. Therefore, sport psychology should focus on the emotional preparation of athletes so that they can manage and control their anxiety at intermediate levels, which in addition to being typical of a competitive situation, is also synonymous with good sporting performance.

## 1. Introduction

The contributions of sport psychology aim to investigate all those variables of a psychological nature that allow us to analyze their impact on sporting performance [1,2], i.e., all those psychological elements and processes experienced by the person practicing sport before, during and after the competition [3,4]. In this regard, the two major psychological constructs that make up the thematic axis of this research correspond to emotional intelligence and sports anxiety, on which both the consequences of their deficit [5] and the benefits of their presence [6] have been investigated.

Regarding emotional intelligence, in this research, we start from the theoretical contributions based on the ability model [7] from which it is defined as a capacity that can be trained, learned, and improved. Therefore, it is understood as a determining psychological skill in the field of sport, which affects both emotional control by athletes [7], as well as decision making, and sporting performance itself [8]. Despite the existence of two major models to explain and understand the concept of emotional intelligence, namely, mixed models and ability models, in this paper, we are committed to the ability model, the research on which has mainly been carried out in the educational field, through standardized questionnaires and taking as a sample mainly adolescents. It is made up of three dimensions or branches: Emotional perception: understood as the ability to resemble and explore one’s own and other people’s feelings. Emotional understanding: related to the ability to specify and probe emotions, retrospectively, both one’s own and others. Emotional regulation: related to the ability to investigate and reason about emotions, both interpersonally and intrapersonal.

However, research on emotional intelligence is not only relegated to this construct, but there is also an increasing prevalence of research focused on the analysis of the relationship established between this construct and other psychological variables, such as adaptation [9,10] or the prevention of risk behaviors [4], burnout syndrome, stress, or anxiety [11,12,13]. We should not forget that numerous studies have shown that low levels of emotional intelligence correspond to the appearance of depression or anxiety [14,15,16,17].

Focusing on anxiety, in the field of sport, a conception of anxiety that refers to a negative emotional state in which the athlete feels a state based on worry, nervousness, tension, and apprehension associated with the activation of bodily arousal has been adopted [18]. Conceptually, there are two ways of approaching the concept of anxiety: trait anxiety and state anxiety [19]. Thus, while the former refers to a trait of the subject’s personality in a stable way, the second refers to a specific situation, which occurs prior to that situation. In view of these contributions, [20] has established itself as a pioneer in speaking of trait anxiety in the field of sport, referring to the anxiety that appears in the athlete in moments prior to or during the competition. In this sense, we would like to highlight the consideration that psychological training in athletes is as important as physical training, since we are referring to a group that may have high levels of anxiety [5,21,22,23,24]. 

Research on anxiety in the field of sport has shown that these emotions appear in pre-competitive situations [25], where the concept of pre-competitive anxiety arises, which can have both a negative and positive effect on the athlete and has been consolidated, being considered within sport psychology as a topic of great interest [26,27,28]. It even manifests itself on a somatic level through excessive30 sweating, trembling of the limbs or increased heart rate, but also on a psychological level, through paralyzing fear, mental dispersion, decreased self-esteem, increased frustration, or guilt [6].

In addition, there are many factors that affect the presence of anxiety, corresponding to the place where the competition takes place, the athlete–coach relationship, the level of performance of the opponent, the expectations placed on the competition, etc. [29,30,31,32,33,34,35], or even when faced with a sporting competition at the beginning or during the competitive process [36,37,38]. The increase in research on this topic has led to the implementation of different techniques by sport psychologists, focusing on teaching coping with fear, visualization, goal setting, relaxation, or improving self-confidence [39,40].

Therefore, the main objective of this research is to analyze the levels of each of the dimensions of emotional intelligence (attention, clarity, and emotional regulation) and their relationship with each of the items related to pre-competitive anxiety that make up the SCAT (Sport Competition). In addition, this research aims: to investigate the influence that one psychological construct has on the other; to investigate the type of relationships that are established between them; and to answer the following research question: Is anxiety an ever-present element in competitive situations, and can sport psychology, specifically, emotional intelligence and emotional training, help the athlete to manage it?

To this end, the following hypotheses are established:

**Hypothesis** **1** **(H1).***The total absence of anxiety does not lead to better sporting performance, and is not related to any of the dimensions of emotional intelligence*. 

**Hypothesis** **2** **(H2).***High levels of anxiety are typical in athletes and are even considered as an emotional drive for sports practice*.

**Hypothesis** **3** **(H3).***Moderate levels of anxiety correspond to good sporting performance, which can be obtained through emotional training*.

## 2. Materials and Methods

### 2.1. Design 

This research uses a quantitative, exploratory, descriptive, and explanatory methodological approach and is based on the survey technique through the use of two validated and standardized questionnaires [41]. The use of this type of design is widely used in fields such as psychology and education to investigate the perception and opinion of the participants as a means of gathering information about a certain phenomenon about which we wish to know their attitude, knowledge, or behaviors [42]. In addition, the sample was selected by convenience, according to the people who attended the class on the day we administered the questionnaires.

Furthermore, it is an ex post facto design because we selected participants whose basic requirement is to be a student of degrees related to physical activity and sport sciences. Thus, students of a degree in physical activity and sport sciences, students of a degree in primary education specializing in physical education, students of a master’s degree in research on physical activity and sport sciences, and students of a master’s degree in secondary education specializing in physical education participated in this research.

### 2.2. Participants

The total sample is made up of 165 students who belong to university degrees (undergraduate and master’s degrees) related to physical activity and sports sciences. Regarding sex, 70.9% (n = 117) were men and 27.9% (n = 46) women, while the mean was 20.33 years, with a standard deviation DE = 3.44. The type of sample used is simple and random. Since the questionnaires were completed by the students who attended class on the day selected to provide them. Therefore, it corresponds to a non-probabilistic and convenience sample. The fundamental reason for choosing students who study degrees related to physical activity and sports sciences is due to the main purpose of this work, in terms of knowledge of the relationship between pre-competitive anxiety and emotional intelligence in university students, future professionals of physical activity and sports sciences. In addition, this work takes an interest in analyzing what knowledge about them they have or need, and how, from the initial training, it is appropriate to work on training in emotions as a protective factor against anxiety in athletes and students. The sample size was calculated using Soper’s a priori sample size calculator for structural models, based on which the recommended minimum effect size was 200 cases, so the total sample of this research is close to the recommended number. Table 1 shows the main socio-demographic characteristics of the sample with respect to age and sex using frequencies and percentages.

As criteria for inclusion of the participants we chose that the participating students were enrolled in the subject and course in question, that they accepted their participation through informed consent (official model of the University of Almería) and that they were of legal age. All participants completed the official informed consent form of the University of Almería (Spain) and were informed of the data protection protocol. On the other hand, as exclusion criteria, we discarded those incomplete booklets.

The N of the universe of the study understood as the set from which the information is extracted was a total of 277 participants, taking as a guide for its calculation: the total number of students enrolled in each of the four courses of the official degree in sciences of physical activity and sport; the total number of students enrolled in the specialty of physical education of the official title of primary education; the total number of students enrolled in the master’s degree in physical activity and sports science; and the total number of students enrolled in the specialty of physical education within the official master’s degree in teacher training. We used the margin of error calculator, and we specified the total population size, the desired confidence level (the standard value used by most researchers is 95%), and the sample size. The score obtained was 3.20%. In this sense, the further the percentage is from 50%, the smaller the margin of error turns out to be.

Figure 1 shows the frequency of the collection, analysis, selection, and inclusion criteria for the total number of questionnaires collected.

### 2.3. Process

The data collection procedure for this research is as follows: 

Phase 1: Design and temporary and theoretical planning of the investigation. Specifically, it is an empirical study based on a quantitative, descriptive, and cross-sectional methodology. 

Phase 2: Our design and planning were submitted to evaluation by the Bioethics Committee of the University of Almería, being approved by the Institutional Review Board of the University of Almería (UALBIO2022/035), given that the Helsinki Declaration guidelines on research ethics are met. 

Phase 3: To access the data collection, we first contacted each of the teachers of each of the undergraduate or master’s degree subjects by email. Through it, we informed them of the reason and purpose of the investigation, and we asked for permission to attend their class in person, and we agreed on a visit date, either at the end or at the beginning of the class.

Phase 4: The collection of information was carried out during the first four-month period of the 2021/2022 academic year in each of the four courses of the official degree in physical activity and sports sciences, in the specialty of physical education of the official title degree of primary education, in the official master’s degree in physical activity and sports sciences, and in the physical education specialty of the official master’s degree in teacher training. 

During the data collection, the principal investigator was always present to invite the participants to participate in the study and obtain their informed consent, as well as offering brief information on the objectives of the research and the pertinent instructions to complete the questionnaire, taking emotional intelligence and anxiety as variables. Before beginning, the anonymity of the answers and the confidentiality of the data were guaranteed, dedicating around 20 min to their answers, without any of them reporting problems to complete it.

### 2.4. Instruments

For this research we created a kind of booklet which was made up of the following instruments: First, different sociodemographic data corresponding to age, sex, degree, course, marital status, or sports practiced were requested. Then, two standardized and validated scales were included in the booklet to measure each of the psychological constructs that articulate the thematic axis of this research, which correspond to the following: To measure pre-competitive anxiety, we used the Sport Competition Anxiety Test (SCAT) [22] adapted to the Spanish population [29]. It corresponds to an instrument used in a very high number of investigations from the 1980s to the present [43]. It is made up of 15 items with three response options on a three-point ordinal scale (almost never, sometimes, or frequently), which makes it possible to measure discrepancies at the individual level in trait anxiety toward competition. The psychometric properties of this instrument are adequate in terms of reliability (Cronbach’s alpha), obtaining a score of 0.78 [44].To measure emotional intelligence, the “TMMS-24” was used as an instrument [44] to measure the dimensions of emotional intelligence: attention, clarity, and regulation, made up of 24 items on a Likert-type scale (1–5). Its psychometric properties are appropriate in terms of reliability (Cronbach’s alpha) (attention: α = 0.92; clarity: α = 0.84; repair: α = 0.84) [30]. In addition, it has high reliability (Cronbach’s alpha) for each dimension (perception, α = 0.90; clarity, α = 0.90; regulation, α = 0.86) and adequate test–retest reliability: perception = 0, 60; comprehension = 0.70 and regulation = 0.83.

Both the hierarchical omega [44] and the explained common variance (ECV) [45] were also calculated. Rank omega scores ≥ 0.70 indicate the presence of a one-dimensional structure [44]. Regarding the second, scores below 0.70 indicate multidimensionality, and values above 0.85 indicate one-dimensionality. Additionally, at the item level, the ECV-I was calculated [44] to know the variance of each item explained by the GFR. In this sense, scores ≥0.80 indicate a significant influence of the GFR [45]. In short, values >0.70 suggest a latent variable adequately defined by its indicators [45].

### 2.5. Data Analysis

Descriptive data analysis involved calculating the mean, standard deviation, and bivariate correlations. In addition, the reliability was calculated, and finally, a second-order structural equation model (SEM) was performed. This statistical test not only demonstrates the higher-level structure and its consequences for certain dependent variables, but also offers us the opportunity to confront irrefutable forms of multicollinearity [46,47,48]. To approve or reject the proposed model, the following indices [36] were taken as reference: TLI (Tucker–Lewis index), SRMR (standardized root mean square residual) and RMSEA (root mean square error of approximation). TLI values above 0.95, SRMR values below 0.06, and RMSEA values below 0.08 were considered adequate. These analyses were performed using SPSS version 28 and R statistical analysis software. Data sets used and/or analyzed during the current study are available from the corresponding author upon reasonable request.

## 3. Results

The results are presented according to each of the dimensions of emotional intelligence, emotional attention, emotional clarity, and emotional regulation. Thus, Table 2 shows the relationships between emotional attention (EA) and each of the 15 items that make up the SCAT. The data are presented based on the scores obtained in the bivariate correlations: the closer to +1, the higher the association. An exact value of +1 would indicate a perfect positive linear relationship. In this case, the variables would be associated in a direct sense, and the correlations between the variables were positive, reflecting reciprocity between the study variables. 

Structural equation model

The hypothetical model of predictive relationships (Figure 2) shows the following indices:

- Overall fit indices (evaluate the model overall) and are adequate: *p* < 0.001, RMSEA = 0.0, GFI = 0.89.

The relationships established in the structural equation model between emotional attention and the SCAT items are specified below: ○Item 1, “competing against others is fun”, was not correlated with emotional attention. ○Item 2, “before competing I feel agitated”, was positively correlated with emotional attention (=0.00, *p* < 0.001). ○Item 3, “before competing I worry about not performing well”, was positively correlated with emotional attention (=0.02, *p* < 0.001). ○Item 4, “I am a good athlete when I compete”, was positively correlated with emotional attention (=0.06, *p* < 0.001). ○Item 5, “when I compete, I worry about making mistakes”, was positively correlated with emotional attention (=0.03, *p* < 0.001). ○Item 6, “before competing I am calm”, was positively correlated with emotional attention (=0.06, *p* < 0.001). ○Item 7, “setting a goal is important when competing”, was positively correlated with emotional attention (=0.01, *p* < 0.001). ○Item 8, “before competing I have an unpleasant feeling in my stomach”, was positively correlated with emotional attention (=0.01, *p* < 0.001). ○Item 10, “I like to compete in physically demanding activities”, was positively correlated with emotional attention (=0.07, *p* < 0.001). ○Item 11, “before competing I feel relaxed”, was positively correlated with emotional attention (=0.05, *p* < 0.001). ○Item 12, “before competing I feel nervous”, was positively correlated with emotional attention (=0.09, *p* < 0.001). ○Item 13, “team sports are more exciting than individual sports”, was positively correlated with emotional attention (=0.01, *p* < 0.001). ○Item 14, “It makes me nervous wanting the competition to start”, was positively correlated with emotional attention (=0.01, *p* < 0.001). ○Item 15, “before competing I usually feel tense”, was positively correlated with emotional attention (=0.02, *p* < 0.001).

After that, Table 3 shows the relationships between emotional clarity (EC), and it is 1 of the 15 items that make up the SCAT. The data are presented on the basis of the scores obtained in the bivariate correlations: the closer to +1, the higher the association. An exact value of +1 would indicate a perfect positive linear relationship. In this case, the variables would be associated in a direct sense, the correlations between the variables were positive, reflecting reciprocity between the study variables. 

Structural equation model

The hypothetical model of predictive relationships (Figure 3) shows the following indices:

- Overall fit indices (evaluate the model overall) and are adequate: *p* < 0.001, RMSEA = 0.0, GFI = 0.89.

The relationships established in the structural equation model between emotional clarity and the SCAT items are specified below: ○Item 1, “competing against others is fun”, did not correlate with the emotional clarity dimensions. ○Item 2, “before competing I feel agitated”, was positively correlated with emotional clarity (=0.02, *p* < 0.001). ○Item 3, “before competing I worry about not performing well”, was positively correlated with emotional clarity (=0.08, *p* < 0.001). ○Item 4, “I am a good athlete when I compete”, was positively correlated with emotional clarity (=0.06, *p* < 0.001). ○Item 5, “When I compete, I worry about making mistakes”, was positively correlated with emotional clarity (=0.03, *p* < 0.001). ○Item 6, “before competing I am calm”, was positively correlated with emotional clarity (=0.06, *p* < 0.001). ○Item 7, “setting a goal is important when competing”, was positively correlated with emotional clarity (=0.01, *p* < 0.001). ○Item 8, “before competing I have an unpleasant feeling in my stomach”, was positively correlated with emotional clarity (=0.01, *p* < 0.001). ○Item 10, “I like to compete in physically demanding activities”, was positively correlated with emotional clarity (=0.07, *p* < 0.001). ○Item 11, “before competing I feel relaxed”, was positively correlated with emotional clarity (=0.05, *p* < 0.001). ○Item 12, “before competing I feel nervous”, was positively correlated with emotional clarity (=0.09, *p* < 0.001). ○Item 13, “team sports are more exciting than individual sports”, was positively correlated with emotional clarity (=0.02, *p* < 0.001). ○Item 14, “It makes me nervous wanting the competition to start”, was positively correlated with emotional clarity (=0.01, *p* < 0.001). ○Item 15, “before competing I usually feel tense”, was positively correlated with emotional clarity (=0.04, *p* < 0.001). 

Finally, Table 4 shows the relationships between emotional regulation (ER) and it is 1 of the 15 items that make up the SCAT. The data are presented on the basis of the scores obtained in the bivariate correlations: the closer to +1, the higher the association. An exact value of +1 would indicate a perfect positive linear relationship. In this case, the variables would be associated in a direct sense, the correlations between the variables were positive, reflecting reciprocity between the study variables. 

Structural equation model

The hypothetical model of predictive relationships (Figure 4) shows the following indices:

- Overall fit indices (evaluate the model overall) and are adequate: *p* < 0.001, RMSEA = 0.0, GFI = 0.89.

The relationships established in the structural equation model between emotional regulation and the SCAT items are specified below: ○Item 1, “competing against others is fun”, did not obtain a correlation with emotional regulation. ○Item 2, “before competing I feel agitated”, was positively correlated with emotional regulation (=0.00, *p* < 0.001). ○Item 3, “before competing I worry about not performing well”, was positively correlated with emotional regulation (=0.02, *p* < 0.001). ○Item 4, “I am a good athlete when I compete”, was positively correlated with emotional regulation (=0.08, *p* < 0.001). ○Item 5, “When I compete, I worry about making mistakes”, was positively correlated with emotional regulation (=0.06, *p* < 0.001). ○Item 6, “before competing I am calm”, was positively correlated with emotional regulation (=0.06, *p* < 0.001). ○Item 7, “setting a goal is important when competing”, was positively correlated with emotional regulation (=0.01, *p* < 0.001). ○Item 8, “before competing I have an unpleasant feeling in my stomach”, was positively correlated with emotional regulation (=0.01, *p* < 0.001). *p* < 0.001). ○Item 10, “I like to compete in physically demanding activities”, was positively correlated with emotional regulation (=0.07, *p* < 0.001). ○Item 11, “before competing I feel relaxed”, was positively correlated with emotional regulation (=0.05, *p* < 0.001). ○Item 12, “before competing I feel nervous”, was positively correlated with emotional regulation (=0.09, *p* < 0.001). ○Item 13, “team sports are more exciting than individual sports”, was positively correlated with emotional regulation (=0.02, *p* < 0.001). ○Item 14, “It makes me nervous wanting the competition to start”, was positively correlated with emotional regulation (=0.01, *p* < 0.001). ○Item 15, “before competing I usually feel tense”, was positively correlated with emotional regulation (=0.06, *p* < 0.001).

Since the SCAT lacks factors created as such, mean scores and their relationship between the total scores obtained on the SCAT and emotional attention, emotional clarity, and emotional regulation was also calculated (see Table 5). The data are presented on the basis of the scores obtained in the bivariate correlations: the closer to +1, the higher the association. An exact value of +1 would indicate a perfect positive linear relationship. In this case, the variables would be associated in a direct sense, the correlations between the variables were positive, reflecting reciprocity between the study variables. 

The results show the direct and positive relationship between SCAT mean scores and emotional attention.

## 4. Discussion

The main objective of this research was to analyze the relationship between the two thematic focuses of this work, that is, between anxiety and emotional intelligence. This objective was met. This fact shows that emotional intelligence refers to a set of skills related to the reflection of emotions and its feasibility to facilitate and govern thought [49]. Additionally, the research response is answered, in that it has been shown that anxiety is an indispensable component of any competitive situation.

Regarding the hypotheses, hypothesis one is fulfilled, and it is demonstrated that the total absence is not related to any of the dimensions of emotional intelligence, which is in line with what has already been provided by other research that establishes that a lack of anxiety is not synonymous with good sports performance [9,50,51]. Regarding hypothesis two, this is also fulfilled, since anxiety is something that the participants state they have and relate to as something of their own prior to the competition, hence the explanation why many elite athletes consider anxiety itself as an impulse to optimize sporting activity [52]. Regarding hypothesis three, it is also fulfilled. In this case, moderate levels of anxiety correspond to good sports performance, which can be obtained through emotional training. This is the main finding of this work, which allows us to affirm that there is a direct and positive relationship between anxiety and emotional intelligence. Therefore, emotional training and sports psychology are the ways to achieve moderate levels of anxiety, which are synonymous with good sports performance [53]. It is precisely here where one of the main actions of sports psychology lies: the emotional preparation towards the athlete to learn to manage and control anxiety to achieve intermediate levels of anxiety [54,55,56]. 

For this reason, the results obtained indicate the importance of betting not only on a physical or cognitive level, but also on an emotional level in the initial training of future professionals in the physical activity and sports sciences. Different implications of an educational nature are derived from this, which go through the promotion of the treatment of emotional intelligence in athletes, based on the theoretical contributions of ability models [9]. 

The heyday and power of research focused on emotional intelligence within the psychological field in the 21st century is undeniable. In this sense, our research shows the importance of studying this construct from the perspective of sport psychology, a field in which its study is also acquiring great importance [57,58,59]. 

In this way, the results provided indicate the close relationship and influence that it has on such important aspects in the field of sport, such as sports performance, training effectiveness, social skills, problem solving, and anxiety [60,61,62]. In this sense, there are numerous investigations that have been carried out in relation to anxiety in participants of different sports: athletes, soccer players, wrestlers, and surfers, among others [63]. 

In addition, another large amount of research has also analyzed the relationship of anxiety with other psychological constructs such as resistance, performance, or the type of training [64,65,66]. In this sense, this research is part of the same thematic line of previously published works related to sports anxiety [31,67,68].

## 5. Limitations and Future Directions

We must be cautious with the findings obtained, due to the sample size of this research and because it corresponds to a cross-sectional research design. Therefore, we consider that it would be appropriate to repeat this investigation, increasing the number of participants to analyze whether these findings are maintained, this being the main limitation of this investigation. Future research topics will focus on the differential analysis according to the type of sport practiced and the time dedicated to the sport. In this way, future research will seek to investigate setbacks according to the degree of professionalization of each sport practiced by future participants, as well as the size of the difference between different groups.

## 6. Conclusions

In conclusion, the present study provides evidence of the relationships hypothesized in the proposed models between the dimensions of emotional intelligence and anxiety. It is shown that anxiety is something of its own in sports practice, mainly before competitions, and it can even be an impulse for the athlete. In turn, emotional intelligence is postulated as a key element in athletes learning to manage these levels of anxiety and being able to maintain them at medium and moderate levels. For this reason, this study has established the importance of providing emotional training to future professionals in physical activity and sports sciences from their initial training, since it will trigger an adequate regulation of their own emotions and anxiety that will allow them to successfully face any sports or work situation.

## Figures and Tables

**Figure 1 ijerph-20-04224-f001:**
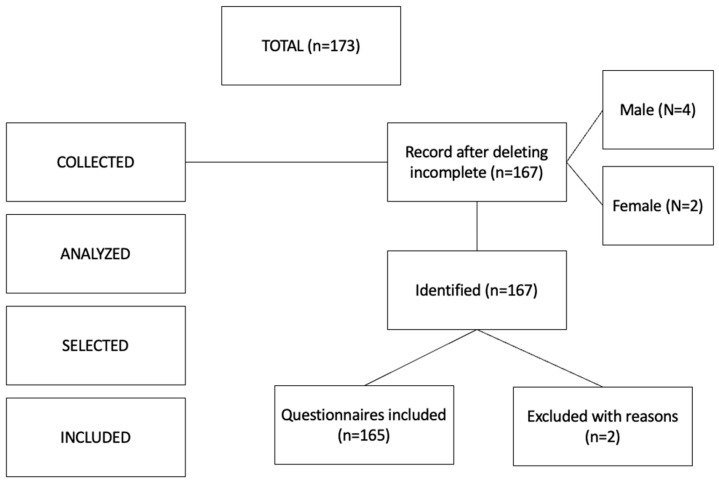
Flowchart of the selection procedure.

**Figure 2 ijerph-20-04224-f002:**
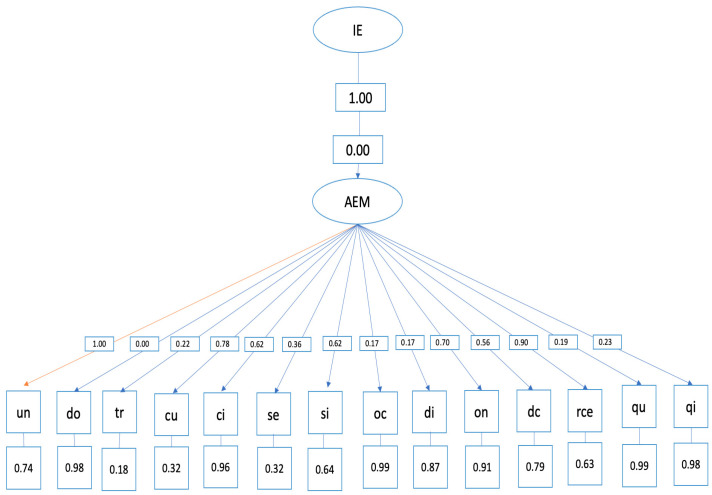
Structural equation model.

**Figure 3 ijerph-20-04224-f003:**
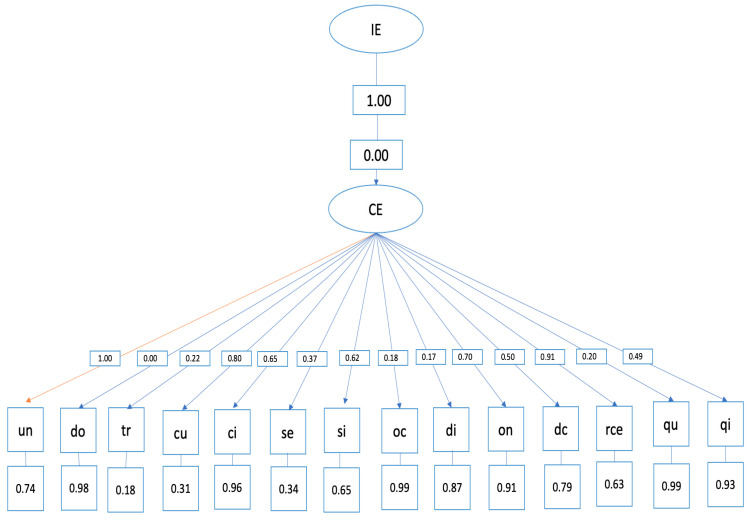
Structural equation model.

**Figure 4 ijerph-20-04224-f004:**
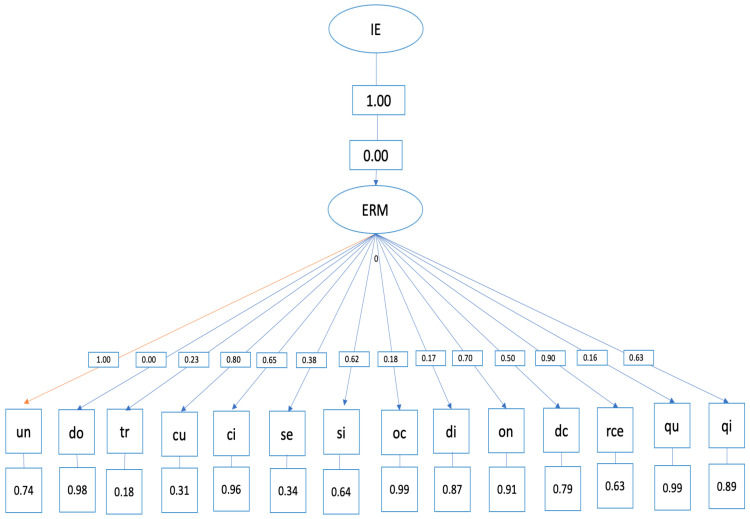
Structural equation model.

**Table 1 ijerph-20-04224-t001:** Description of the sample according to age and sex.

	Females	Males	Total	Under 25 Years	Over 25 Years
First course	18 (39.1%)	68 (58.1%)	86	86 (97.8%)	2 (2.2%)
Second course	16 (34.8%)	23 (19.7%)	39	38 (97.4%)	1 (2.6%)
Third year	6 (13%)	14 (12%)	20	20 (100%)	0
Total	40	105	145		
Master’s degree	6 (13%)	12 (10.3%)	18	4 (22.3%)	17 (77.7%)
Total	46	117	163		

**Table 2 ijerph-20-04224-t002:** Preliminary analyses.

	AE	1	2	3	4	5	6	7	8	9	10	11	12	13	14	15
AE		0.057	0.098	0.204 **	0.085	0.189 *	0.057	0.222 **	0.090	0.165 *	0.064	0.019	0.058	0.061	0.178 *	0.124
1			0.031	0.094	0.254 **	0.079	0.013	0.074	0.111	0.001	0.156 *	0.066	0.089	0.003	0.026	0.097
2				0.448 **	0.186 *	0.253 **	0.501 **	0.091	0.435 **	0.425 **	0.142	0.505 **	0.541 **	0.106	0.336 **	0.417 **
3					0.181 *	0.336 **	0.325 **	0.056	0.413 **	0.377 **	0.056	0.339 **	0.356 **	0.009	0.303 **	0.436 **
4						0.033	0.313 **	0.030	0.294 **	0.216 **	0.358 **	0.153	0.293 **	0.088	0.007	0.207 **
5							0.219 **	0.058	0.192 *	0.291 **	0.073	0.277 **	0.314 **	0.064	0.180 *	0.195 *
6								0.012	0.480 **	0.555 **	0.186 *	0.705 **	0.681 **	0.115	0.300 **	0.442 **
7									0.003	0.007	0.163 *	0.094	0.086	0.196 *	0.107	0.102
8										0.567 **	0.216 **	0.446 **	0.482 **	0.106	0.327 **	0.520 **
9											0.125	0.652 **	0.743 **	0.131	0.548 **	0.624 **
10												0.125	0.154 *	0.018	0.128	0.185 *
11													0.793 **	0.048	0.345 **	0.498 **
12														00.92	0.429 **	0.644 **
13															0.109	0.075
14																0.519 **
15																

Note. * *p* < 0.05; ** *p* < 0.01.

**Table 3 ijerph-20-04224-t003:** Preliminary analyses.

	CE	1	2	3	4	5	6	7	8	9	10	11	12	13	14	15
CE		0.020	0.200 *	0.193 *	0.101	0.039	0.179 *	0.082	0.111	0.105	0.166 *	0.277 **	0.223 **	0.027	0.138	0.112
1			0.031	0.094	0.254 **	0.079	0.013	0.074	0.111	0.001	0.156 *	0.066	0.089	0.003	0.026	0.097
2				0.448 **	0.186 *	0.253 **	0.501 **	0.091	0.435 **	0.425 **	0.142	0.505 **	0.541 **	0.106	0.336 **	0.417 **
3					0.181 *	0.336 **	0.325 **	0.056	0.413 **	0.377 **	0.056	0.339 **	0.356 **	0.009	0.303 **	0.436 **
4						0.033	0.313 **	0.030	0.294 **	0.216 **	0.358 **	0.153	0.293 **	0.088	0.007	0.207 **
5							0.219 **	0.058	0.192 *	0.291 **	0.073	0.277 **	0.314 **	0.064	0.180 *	0.195 *
6								0.012	0.480 **	0.555 **	0.186 *	0.705 **	0.681 **	0.115	0.300 **	0.442 **
7									0.003	0.007	0.163 *	0.094	0.086	0.196 *	0.107	0.102
8										0.567 **	0.216 **	0.446 **	0.482 **	0.106	0.327 **	0.520 **
9											0.125	0.652 **	0.743 **	0.131	0.548 **	0.624 **
10												0.125	0.154 *	0.018	0.128	0.185 *
11													0.793 **	0.048	0.345 **	0.498 **
12														00.92	0.429 **	0.644 **
13															0.109	0.075
14																0.519 **
15																

Note. * *p* < 0.05; ** *p* < 0.01.

**Table 4 ijerph-20-04224-t004:** Preliminary analyses.

	ER	1	2	3	4	5	6	7	8	9	10	11	12	13	14	15
ER		0.010	0.175 *	0.184 *	0.156 *	0.048	0.284 **	0.079	0.272 **	0.181 *	0.284 **	0.338 **	0.272 **	0.111	0.015	0.258 **
1			0.031	0.094	0.254 **	0.079	0.013	0.074	0.111	0.001	0.156 *	0.066	0.089	0.003	0.026	0.097
2				0.448 **	0.186 *	0.253 **	0.501 **	0.091	0.435 **	0.425 **	0.142	0.505 **	0.541 **	0.106	0.336 **	0.417 **
3					0.181 *	0.336 **	0.325 **	0.056	0.413 **	0.377 **	0.056	0.339 **	0.356 **	0.009	0.303 **	0.436 **
4						0.033	0.313 **	0.030	0.294 **	0.216 **	0.358 **	0.153	0.293 **	0.088	0.007	0.207 **
5							0.219 **	0.058	0.192 *	0.291 **	0.073	0.277 **	0.314 **	0.064	0.180 *	0.195 *
6								0.012	0.480 **	0.555 **	0.186 *	0.705 **	0.681 **	0.115	0.300 **	0.442 **
7									0.003	0.007	0.163 *	0.094	0.086	0.196 *	0.107	0.102
8										0.567 **	0.216 **	0.446 **	0.482 **	0.106	0.327 **	0.520 **
9											0.125	0.652 **	0.743 **	0.131	0.548 **	0.624 **
10												0.125	0.154 *	0.018	0.128	0.185 *
11													0.793 **	0.048	0.345 **	0.498 **
12														00.92	0.429 **	0.644 **
13															0.109	0.075
14																0.519 **
15																

Note. * *p* < 0.05; ** *p* < 0.01.

**Table 5 ijerph-20-04224-t005:** Preliminary analyses.

	SCAT	AE	CE	RE
SCAT		0.321 **	−0.069	−0.052
AE			0.127	0.263 **
CE				0.508 **
RE				

Note. ** *p* < 0.01.

## Data Availability

The data is not publicly available due to ethical or privacy restrictions, however, it may be available if requested from the corresponding author.

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
