# Peer review of "The Importance of Emotional Intelligence in University Athletes: Analysis of Its Relationship with Anxiety"

_ijerph, 2023, doi:10.3390/ijerph20054224_

Round 1

Reviewer 1 Report (Previous Reviewer 2)

Great work!
However, you should send the files of graphs in a minimum of 300 DPI.

Best wishes

Author Response

Good job! However, you must send the graphics files in a minimum of 300 DPI. Best wishes

In order to improve the visualization of both you and the editor, attached is a file with the models of equations both in Word format, as in pdf format, as  in jpg format  so that the journal decides which format to use in case of being accepted.

Reviewer 2 Report (Previous Reviewer 1)

The manuscript remains with many limitations.
Basic aspects: the abstract is not between 250-300 words;
The hypotheses should all be based on the literature that can support them (insufficient information).
Study design is not explained, only identified.
Tables have no prior contextualization;
The values in the tables (table 1 ) are mixed, absolute values with percentages without indications.
For example, table 2, 3, 4 readers have to think about whether they are reading correlation values, levels or significance values.
I don't understand table 5. There are no clear explanations of the information.
In the structural equations model there are still non-visible numbers. The items to which each value corresponds are not clear.
Discussion and conclusion, very, very incomplete.

Author Response

The manuscript remains with many limitations.

Dear reviewer, thank you for your time, your dedication and your contributions that undoubtedly help to improve this research. Below we respond to their suggestions (in the manuscript they are marked in blue).

Basics: the abstract is not between 250-300 words;

  • The summary has been modified and expanded. It currently contains a total of 278 words (lines 8-24).

All hypotheses should be based on the literature that can support them (insufficient information).

  • The hypotheses have been reformulated according to the existing contributions on the subject from the scientific literature and research carried out in the field of sports psychology (lines 117-122).

The design of the study is not explained, it is only identified.

  • The design used has been explained, indicating that a quantitative, exploratory, descriptive and explanatory methodological approach is used and the survey technique is based on the use of two validated and standardized questionnaires. The use of this type of design is widely extended in areas such as psychology and education in order to investigate the perception and opinion of the participants as a means of collecting information before a certain phenomenon about which we want to know their attitude, knowledge or behavior.  In addition, the selection of the sample was of type no and for convenience, according to the people who attended class on the day we provided the questionnaires. In addition, it is an ex post facto design because we select participants whose basic requirement is to be a student of degrees related to physical activity and sport sciences. In this way, students of the degree in physical activity and sports sciences, students of the degree of primary education, mention in physical education, students of the master's degree in research in physical activity and sports sciences, and students of the master's degree in secondary education specializing in physical education  (lines 130-144) participated in this research.

Tables have no prior contextualization;

  • Prior to each of the tables that appear in the manuscript, a brief information about the data they show has been added. Table 1 (lines 161-163), table 2 (lines 283-290/ 292), table 3 (lines 345-349/ 351), table 4 (lines 397-401/ 403)

The values in the tables (Table 1) are mixed, absolute values with percentages without indications.

  • It has been indicated that the scores shown in the table refer to percentages and frequencies. And the indication of the percentage symbol has been added in those scores that correspond to them (lines 161-163).

For example, readers of tables 2, 3, 4 have to think about whether they are reading correlation values, levels, or significant values.

  • In tables 2, 3 and 4 it has been previously added that the data  contained in the tables are obtained  from the scores obtained in the bivariate correlations. The closer to +1, the higher your association. An exact value of +1 would indicate a perfect positive linear relationship. And, that in this case the variables would be directly associated, the correlations between the variables were positive, reflecting the reciprocity between the study variables. TAbla 2 (lines 283-290/ 292), Table 3 (lines 345-349/ 351), Table 4 (lines 397-401/ 403)

I don't understand Table 5. There are no clear explanations of the information.

  • A brief explanation and justification of the elaboration and incorporation of Table 5 has been added, Since the SCAT lacks factors created as such, the mean scores and their relationship between the total scores obtained in the SCAT and emotional attention, emotional clarity and emotional regulation were calculated. Data are presented based on scores obtained in bivariate correlations. The closer to +1, the greater the association. An exact value of +1 would indicate a perfect positive linear relationship. In this case the variables would be directly associated, the correlations between the variables were positive, reflecting reciprocity between the study variables (lines 447-453/456).

In the structural equation model there are still non-visible numbers.  The elements to which each value corresponds are unclear.

  • In order to improve the visualization of both you and the editor, attached is a file with the models of equations both in Word format, as in pdf format, as  in jpg format  so that the journal decides which format to use in case of being accepted.

Discussion and conclusion, very, very incomplete.

  • The discussion has been rewritten. It addresses contents such as the main objective, the research question, the fulfillment of the hypotheses, the main findings, the educational implications of the results and other theoretical contributions related to our research. A total of 18 new references have been added to this section (lines 461-504).
  • The conclusions section has also been rewritten (lines 517-526)

This manuscript is a resubmission of an earlier submission. The following is a list of the peer review reports and author responses from that submission.

Round 1

Reviewer 1 Report

My suggestions for authors are:

Abstract: The abstract is very incomplete and must be well structured;

Introduction: The variables must be well defined; The authors must indicate the theoretical models of reference; There must be a better connection of ideas in the introduction and the results of the studies used must be clear; The authors should provide much more information about the dimensions of the study variables that are the target of analysis; It is not necessary to write about measuring instruments, it is not the objective of the study; Authors should indicate only one research question and duly substantiated; Each hypothesis must be theoretically substantiated, which does not happen. All acronyms must be defined.

Method: Figure 1 is not objective nor is it previously explained. There is no procedure.

Results: The results and figures are not clear. There are numbers that I can't see.

Discussion and Conclusions: very incomplete.
